# Overexpression of Egr1 Transcription Regulator Contributes to Schwann Cell Differentiation Defects in Neural Crest-Specific *Adar1* Knockout Mice

**DOI:** 10.3390/cells13231952

**Published:** 2024-11-23

**Authors:** Lisa Zerad, Nadjet Gacem, Fanny Gayda, Lucie Day, Ketty Sinigaglia, Laurence Richard, Melanie Parisot, Nicolas Cagnard, Stephane Mathis, Christine Bole-Feysot, Mary A. O’Connell, Veronique Pingault, Emilie Dambroise, Liam P. Keegan, Jean Michel Vallat, Nadege Bondurand

**Affiliations:** 1Laboratory of Embryology and Genetics of Human Malformations, Imagine Institute, INSERM UMR 1163, Université Paris Cité, 24 Boulevard du Montparnasse, 75015 Paris, France; lisa.zerad@institutimagine.org (L.Z.); nadjet.gacem@inserm.fr (N.G.); fanny.gayda@institutimagine.org (F.G.); lucie.day@institutimagine.org (L.D.); veronique.pingault@inserm.fr (V.P.); 2Central European Institute for Technology, Masaryk University (CEITEC MU), Kamenice 735/5, 625 00 Brno, Czech Republic; ketty.sinigaglia@ceitec.muni.cz (K.S.); mary.oconnell@ceitec.muni.cz (M.A.O.); liam.keegan@ceitec.muni.cz (L.P.K.); 3Department of Neurology, Centre de Reference “Neuropathies Périphériques Rares”, CHU Limoges, 87000 Limoges, France; laurence.richard@cegetel.net (L.R.); jean-michel.vallat@unilim.fr (J.M.V.); 4Genomics Core Facility, Institut Imagine-Structure Fédérative de Recherche Necker, INSERM U1163 et INSERM US24/CNRS UAR3633, Paris Descartes Sorbonne Paris Cite University, 75015 Paris, France; melanie.parisot@institutimagine.org (M.P.); christine.bole@inserm.fr (C.B.-F.); 5Bioinformatics Platform, Imagine Institute, INSERM UMR 1163, 75015 Paris, France; nicolas.cagnard@yahoo.fr; 6Department of Neurology (Nerve-Muscle Unit) and ‘Grand Sud-Ouest’ National Reference Center for Neuromuscular Disorders, CHU Bordeaux, Pellegrin Hospital, 33000 Bordeaux, France; stephane.mathis@chu-bordeaux.fr; 7Laboratory of Molecular and Physiopathological Bases of Osteochondrodysplasia, Imagine Institute, INSERM UMR 1163, Université Paris Cité, 24 Boulevard du Montparnasse, 75015 Paris, France; emilie.dambroise@institutimagine.org

**Keywords:** Schwann cells, differentiation, ADAR1, MAVS, EGR1, neural crest

## Abstract

Adenosine deaminase acting on RNA 1 (ADAR1) is the principal enzyme for the adenosine-to-inosine RNA editing that prevents the aberrant activation of cytosolic nucleic acid sensors by endogenous double stranded RNAs and the activation of interferon-stimulated genes. In mice, the conditional neural crest deletion of *Adar1* reduces the survival of melanocytes and alters the differentiation of Schwann cells that fail to myelinate nerve fibers in the peripheral nervous system. These myelination defects are partially rescued upon the concomitant removal of the Mda5 antiviral dsRNA sensor in vitro, suggesting implication of the Mda5/Mavs pathway and downstream effectors in the genesis of *Adar1* mutant phenotypes. By analyzing RNA-Seq data from the sciatic nerves of mouse pups after conditional neural crest deletion of *Adar1* (*Adar1*cKO), we here identified the transcription factors deregulated in *Adar1*cKO mutants compared to the controls. Through *Adar1*;*Mavs* and *Adar1*cKO;*Egr1* double-mutant mouse rescue analyses, we then highlighted that the aberrant activation of the Mavs adapter protein and overexpression of the early growth response 1 (EGR1) transcription factor contribute to the *Adar1* deletion associated defects in Schwann cell development in vivo. In silico and in vitro gene regulation studies additionally suggested that EGR1 might mediate this inhibitory effect through the aberrant regulation of EGR2-regulated myelin genes. We thus demonstrate the role of the Mda5/Mavs pathway, but also that of the Schwann cell transcription factors in *Adar1-*associated peripheral myelination defects.

## 1. Introduction

Adenosine deaminase acting on RNA 1 (ADAR1) catalyzes adenosine-to-inosine deamination in double-stranded RNA molecules to regulate cellular responses to endogenous and exogenous RNA [1,2,3,4]. In mice, *Adar1* deletion is lethal by embryonic day (E)12.5 due to fetal liver disintegration and failure of hematopoietic progenitor survival and maturation. In humans, monoallelic pathogenic variations of *ADAR1* cause dyschromatosis symmetrica hereditaria (DSH: MIM **#** 127400, characterized by hyper- and hypo-pigmented macules on the extremities that appear in infancy) [5,6], and biallelic variations or a specific dominant-negative substitution in *ADAR1* result in a spectrum of neuroinflammatory phenotypes including Aicardi-Goutières syndrome (AGS, MIM# 615010), a genetically determined inflammatory encephalopathy ascribed to the group of type 1 interferonopathies [7,8,9,10,11]. Clinical and genetic studies have helped to define a cell-intrinsic mechanism for the initiation of autoinflammation/autoimmunity upon human *ADAR1* or mouse *Adar1* deficiency: without appropriate RNA editing by ADAR1, endogenous dsRNA aberrantly activates the antiviral cytosolic dsRNA-sensing receptor MDA5, leading to the recruitment of mitochondrial antiviral-signaling (MAVS) adapter protein and to the inappropriate activation of type 1 interferon (IFN) expression, the upregulation of IFN-stimulated genes (ISGs), and cell death [1,2,4,11]. Consistent with this model, the embryonic lethality of *Adar1* mutant mice is rescued upon the simultaneous knock-out of either *Ifih1* (encoding Mda5) or *Mavs* [12,13,14]. The contributions of additional sensors have more recently been highlighted in both mouse and human [15,16,17,18,19,20]. Independent of the A-to-I editing activity of ADAR1, it involves the competition for dsRNA binding between ADAR1 and protein kinase R (PKR), and between ADAR1 and Z DNA-binding protein 1 (ZBP1). Beyond dsRNA sensors, the description of transcriptional regulators involved in *ADAR1*-mutant developmental alterations is thus far limited (for review, see [21,22,23,24]).

Recently, *ADAR1* has been shown to be required for the regulation of the survival and differentiation of at least two cell types derived from the neural crest (NC), a transient embryonic structure that gives rise to a wide variety of cell types in vertebrates [25,26,27,28]. Indeed, the tissue-specific deletion of *Adar1* in mouse NC cells, making use of the HtPA-Cre mice expressing Cre recombinase under the control of the human tissue plasminogen activator promoter, leads to skin depigmentation, which was shown to result from the Mda5-dependent apoptosis of NC-derived melanocytes at embryonic day (E)18.5 [27]. Defects in myelinating-Schwann cells were also observed.

Schwann-cell precursors represent a nerve-associated embryonic cell type that spread throughout the body using peripheral nerves as navigational scaffolds during vertebrate embryonic development [29]. Being multipotent, these cells can detach from the nerves in specific locations and produce large quantities of pigment cells, autonomic and enteric neurons, chromaffin cells of the adrenal medulla, and specific mesenchymal populations at cranial locations [29]. When in contact with peripheral nerves, and under the control of external stimuli and transcriptional regulators that form well-defined regulatory networks (including positive regulators such as Oct6, Sox10, and Egr2 and myelin repressors such as Tfap2a, and Egr1), these cells develop into immature Schwann cells and then into pro-myelinating Schwann cells, which establish one-to-one relationships with large caliber axons, before finally transforming into myelin-forming mature Schwann cells [30,31,32,33,34,35,36]. Other non-myelinating Schwann cells stay in contact with groups of small-caliber axons and form the so-called Remak bundles. In NC-specific *Adar1*cKO mice, stalling of Schwann cell development at the pro-myelinating stage, without further evidence of cell degeneration, was found from E18.5 onwards [27]. To understand the mechanisms underlying these alterations, a transcriptomic analysis (RNA-Seq) was previously performed on RNA from the sciatic nerves of the *Adar1*cKO mutant versus the control animals at post-natal day 4. It revealed 3009 differentially expressed mRNAs (fold change FC ≥ 2) between the two genotypes and showed that a large number of transcripts upregulated in mutants are associated with IFN activation/viral infection (approximately 75% including *Ifih1*, the deletion of which partially rescued *Adar1*cKO Schwann cell defects in vitro [27]). In addition, 346 transcriptional regulators were deregulated by more than twofold in mutants relative to the controls, but the timing of their expression/deregulation, their connections with the Mda5/Mavs signaling pathway, and their roles in the genesis of myelin defects observed in *Adar1*cKO animals were not determined.

Through a combined analysis of previously published and newly generated sciatic nerve RNA-Seq datasets, we refined the list of transcriptional regulators that were deregulated in the sciatic nerves of *Adar1*cKO mutants relative to the controls and characterized the timing of the expression/deregulation of the most relevant candidates. The rescue of the myelination process, along with the expression profiles of the earliest deregulated transcription factors, were then analyzed in conventional *Adar1;Mavs* double mutants. We additionally focused on the contribution of the early growth response-1 (Egr1) transcription factor to the ADAR1-mutant defects in the regulation of Schwann-cell development in vivo. Overall, our results highlight that overexpression of the Egr1 transcription factor induced by activation of the Mda5/Mavs pathway contributes to the defects in Schwann-cell myelination observed upon the NC-specific deletion of *Adar1* in mice. We also show that EGR1 might mediate this effect by the aberrant regulation of several EGR2-regulated myelin genes.

## 2. Materials and Methods

### 2.1. Mice, Genotyping, Tissue Collection, and Histology

*Adar^fl/fl^* (B6.129-*Adar*^tm1knk^/Mmjax; Jackson laboratory stock number 34619-JAX) was crossed with HtPA-Cre (Tg(PLAT-Cre)116Sdu [37]) to generate HtPA-Cre*:Adar1^fl/+^* animals. The latter were then backcrossed with *Adar1^fl/fl^* to generate progenies of different genotypes: HtPA-Cre; *Adar1^fl/fl^* referred as *Adar1*cKO mutants; wild-type *Adar1^fl/fl^*; or *Adar1^fl/+^* and heterozygous for *Adar1* in NC (HtPA-Cre; *Adar1^fl/+^*), the latter three are referred to as controls.

Alternatively, *Adar^fl/fl^* and HtPA-Cre;*Adar1^fl/+^* were crossed with *Egr1^+/LacZ^* (B6;129-Egr1tm1Pch/Orl [38], also named Krox24 lacZ, RRID:MGI:6334565) to generate HtPA-Cre;*Adar1^fl/+^*; *Egr1^+/LacZ^* and *Adar^fl/fl^*; *Egr1^+/LacZ^
*animals (two generations), which were then intercrossed to generate progenies of different genotypes including *Adar1^fl/+^*; *Egr1^LacZ/LacZ^* or *Adar1^fl/fl^*; *Egr1^LacZ/LacZ^* and HtPA-Cre; *Adar1^fl/fl^* considered as *Egr1* and *Adar1*cKO single mutants, respectively; HtPA-Cre; *Adar1^fl/fl^*; *Egr1^LacZ/+^* (removal of one *Egr1* allele in *Adar1*cKO, i/e *Adar1*cKO *Egr1*Het) and HtPA-Cre; *Adar1^fl/fl^*; *Egr1^LacZ/LacZ^* double mutants (referred as *Adar1*cKO; *Egr1* DM).

In parallel, *Adar^Δ2−13/+^*; *Mavs^−/−^* animals were intercrossed to generate *Adar^Δ2−13/Δ2−13^*; *Mavs^−/−^* double mutants (referred to as *Adar1;Mavs* DM) and *Adar^Δ2−13/+^; Mavs^−/−^ or Adar1^+/+^; Mavs^−/−^* (referred to as the controls) [13].

In each case, tail biopsies were used to perform DNA extraction, making use of the direct PCR lysis reagent for mouse tail (Viagen, Biotech Inc., Los Angeles, CA, USA) and subsequent genotyping. All primers are available upon request.

For electron microscopy, mutant and control legs or dissected sciatic nerve samples from newborn, P4, and P14 animals were collected, fixed, and sectioned, as described in [27].

Alternatively, sciatic nerves from P4 or newborn animals and E17.5 and E18.5 embryos of different genotypes were dissected and used for immunohistochemistry or RT-qPCR experiments.

For histology, heads of E13.5 *Adar1*cKO;*Egr1* DM and control embryos were collected, fixed overnight at 4 °C in 4% paraformaldehyde, and embedded in paraffin. Sagittal sections (5 μm) were stained with hematoxylin-eosin reagent using the standard protocol. Slides were scanned by making use of a slide scanner and visualized using NDPview2 software (version 2.9.29). 

### 2.2. Mouse Schwann Cell Culture and Transfection

Mouse Schwann cells (MSCs, ScienCell, San Diego, CA, USA) were cultured in Schwann cell medium (SCM, ScienCell). Twenty-four hours prior to transfection, the cells were divided and plated at a density of 5 × 10^4^ cells on 24-well plates coated with poly-L-lysine (Sigma-Aldrich, St. Louis, MO, USA) and maintained in SCM. Cells were transfected with the mouse pcDNA3-Egr1 plasmid (Addgene, Watertown, MA, USA) using the lipofectamine LTX with PLUS Reagent (Invitrogen, Thermo Fisher Scientific, Waltham, MA, USA) following the manufacturer’s instructions. Forty-eight hours later, the cells were collected to perform RNA extraction and RT-qPCR as described below.

### 2.3. Mixed Schwann Cells and Neuron Primary Culture

The isolated dorsal root ganglia (DRG) of E13.5 embryos (issued from *Adar1*cKO, *Adar1*cKO *Egr1*Het or *Adar1*cKO;*Egr1* DM and respective controls or simple mutants) were used to perform mixed Schwann cells/sensory neurons cultures as previously described [27]. At the end of culture, cells were fixed in 4% paraformaldehyde (PFA) before being used for immunocytochemistry. Alternatively, cells were collected to perform RNA extraction and RT-qPCR.

### 2.4. RNA Extraction and Quantitative Real-Time PCR

Total RNA from sciatic nerves at different stages, from the mixed primary cultures of *Adar1*cKO, *Adar1*cKO;*Egr1Het*, *Adar1*cKO;*Egr1* DM, *Adar1;Mavs* DM, and the respective controls from MSC cultures were extracted using the RNeasy Mini or Micro Kit (Qiagen, Germantown, MD, USA) according to the manufacturer’s instructions. After DNAse treatment, the concentration and quality of the RNA was determined using NanoDrop (ThermoFisher Scientific) and Xpose (Trinean, Gent, Belgium) apparatus. cDNA was synthesized using Maxima First Strand cDNA Synthesis. Quantitative real-time PCR (qPCR) was performed using the Maxima SYBER green/ROX qPCR Master Mix X2 (all from Thermofisher) and amplified using Mastercycler^®^ RealPlex^2^ (Eppendorf, Hamburg, Germany). The PCR settings were 95 °C for 10 min, then 40 cycles of denaturation at 95 °C for 15 s, followed by annealing and extension at 60 °C for 1 min. The relative abundance values of each amplification product were normalized to the internal control β-actin, and the mRNA expression levels in mutants expressed relative to the controls. The PCR primers are reported in [27] and Appendix A.

### 2.5. Immunocytochemistry

Immunostaining was performed as previously described [27]. Briefly, cells or E18.5, P0, or P4 sciatic nerve sections were immunostained with Tuj1 (MMS-435P, lot TU17044, mouse, 1:1000 dilution, Eurogentec, Seraing, Belgium and ab52623, Rabbit, 1:500 dilution, Abcam, Eugene, OR, USA), Mbp (ab40390, Rabbit, 1:200 dilution, Abcam), anti-EGR1 (15F7, 4153S, Rabbit, 1:750, Cell Signaling Technology, Inc., Danvers, MA, USA), and anti-TFAP2a (Santa Cruz, Dallas, TX, USA sc12726, Mouse, 1:50 and Sigma, SAB4502948, Rabbit, 1:400). Secondary antibodies used were as follows: anti-mouse alexaFluor 568, anti-mouse alexaFluor 488, and anti-rabbit alexaFluor 555 (1:500 dilution, Invitrogen). Preparations were then mounted using Vectashield containing DAPI (Vector laboratories, Newark, CA, USA) and observed using Spinning Disk or Axiovert A1 from Zeiss (Zeiss Microscopy, Cambridge, United Kingdom). Alternatively, slides were scanned making use of a slide scanner and visualized using NDPview2 software.

### 2.6. RNA-Seq Library Preparation, Sequencing, and Bioinformatics Analysis

Total RNA of *Adar1*cKO or the control sciatic nerves were isolated using the RNeasy Kit (Qiagen) including a DNAse treatment step. RNA quality was assessed by capillary electrophoresis using RNA Screen Tape 6000 Pico LabChips with Tape Station (Agilent Technologies, Palo Alto, CA, USA), and the RNA concentration was measured by spectrophotometry using Xpose (Trinean). The Ovation Mouse RNA-Seq System from NuGEN was used to prepare the RNA-Seq libraries from 50 ng of total RNA as recommended by the manufacturer. This kit performs strand-specific RNA-Seq library construction using from 10 to 100 ng of total RNA and uses insert dependent adaptor cleavage (InDA-C) technology to remove the ribosomal RNA transcripts. The Ovation Universal RNA-Seq System provides 16 unique barcoded adaptors to enable multiplex sequencing. To ensure the removal of any DNA contamination that may increase the percentage of intergenic reads, a DNase treatment with HL-dsDNase (ArcticZymes, Tromsø, Norway) was performed prior to the reverse transcription. After the reverse transcription of 50 ng of total RNA and second strand synthesis, a fragmentation step was performed before Illumina compatible adapter ligation. The ligation was followed by the strand selection enzymatic reaction to keep the information about the sense of the transcripts. Insert dependent adaptor cleavage (InDA-C) specific primers were then used to target the depletion of human ribosomal RNA sequencing transcripts before PCR enrichment. To ensure that no excess of amplification was performed during the final PCR step, the number of PCR cycles applied to each sample was evaluated in a preliminary qPCR test using EvaGreen (Biotium, Fremont, CA, USA).

Equimolar pools of the final indexed RNA-Seq libraries were sequenced on an Illumina HiSeq 2500 (Illumina Inc., San Diego, CA, USA) (paired-end reads 130 bases + 130 bases, 12 libraries per lane of FlowCell) to achieve a sequencing depth of ~30 millions paired-end reads per library.

FASTQ files were mapped to the ENSEMBL Mouse GRCm38/mm10 reference using HISAT2 and counted by featureCounts from the Subread R package. Read count normalizations and group comparisons were performed by three independent and complementary statistical methods: Deseq2, edgeR, and LimmaVoom. Flags were computed from counts normalized to the mean coverage. All normalized read counts < 20 were considered as the background (flag 0) and ≥20 as the signal (flag = 1). P50 lists used for the statistical analysis regrouped the genes showing flag = 1 for at least half of the compared samples. The results of the three methods were filtered at a *p* value ≤ 0.05 and twofold or fivefold compared and grouped by Venn diagram. The data were processed making use of R Project for Statistical Computing [http://www.r-project.org/ (R version 3.6.1 (accessed on 5 July 2019))] and Java Treeview-extensible for visualization of the microarray data [39].

The RNA-Seq data have been submitted to the Annotare repository and accepted under the accession number ArrayExpress E-MTAB-11197.

### 2.7. Electronic Microscopy Quantifications

The Schwann-cell count was performed manually on photographs of nerve biopsies (semi-thin sections and/or EM images) of *Adar1;Mavs* DM and HtPA-Cre; *Adar1^fl/fl^; Egr1^LacZ/+^* (*Adar1*cKO;*Egr1*Het), *Egr1^LacZ/LacZ^*, HtPA-Cre; *Adar1^fl/fl^* mutant mice and their respective age-matched controls, and reported to the surface studied to determine a density of cells per square millimeter (mm^2^). After having identified and counted the myelinated nerve fibers (and nerve fibers in the process of myelination) and Remak bundles on EM pictures, the size of the axons was measured using ImageJ freeware (National Institutes of Health, Bethesda, MD, USA), along with the g-ratio as described in [40].

### 2.8. Statistical Analyses

All graphs and statistical analyses were conducted using GraphPad Prism 7.0. Data were presented as the mean ± SEM or SD in the graphs and analyzed using the unpaired two-tailed Student’s *t*-test for simple comparison or one way ANOVA for multiple comparisons. A *p* < 0.05 was considered statistically significant. The *p*-values were as follows: not significant (ns) > 0.05, * <0.05, ** <0.01, *** <0.001, and **** <0.0001.

## 3. Results

### 3.1. Identification of Transcriptional Regulators Showing Aberrant Expression in the Sciatic Nerves of Adar1cKO Mutants by RNA-Seq and In Silico Analyses

To determine whether transcriptional regulators could contribute to the genesis of the Schwann-cell alterations observed in *Adar1*cKO mice, we used previously published (RNA-Seq 1, three controls versus three mutants, [27]) combined with a new set of data generated from RNA extracted from the sciatic nerves of an additional four *Adar1*cKO mice versus four control animals at P4 (RNA-Seq 2). From both datasets, we selected differentially expressed transcripts encoding proteins associated with transcriptional regulation GO terms [positive and negative regulation of gene expression, positive or negative regulation of transcription from RNA polymerase II promoter, transcription factor activity/transcriptional repressor activity, RNA polymerase II core promoter proximal region sequence-specific binding/sequence-specific DNA binding, regulatory region DNA binding, transcription factor binding, negative and positive regulation of transcription, and DNA-templated]. We thus identified 180 transcripts encoding transcriptional regulators that were similarly up- or downregulated with an FC ≥ 2 in the two datasets (Appendix A). Among these, 52 were deregulated with an FC ≥ 5 in *Adar1*cKO mutants relative to the controls (Figure 1 and Appendix A).

We first searched for the most promising/relevant candidates using: (i) the literature, with the keywords “neural crest” and “Schwann cells” to identify the genes with known functions in these cell types of interest, and (ii) various databases to obtain information on subcellular localization and/or cell or non-cell autonomous functions/modes of action of the encoded proteins (Appendix A). These analyses showed that 53.8% (28/52) had a previously reported function in one or other of these cell types (Figure 1 and Appendix A). Twenty-four encoded membrane-associated proteins, cytokines, or neurotrophic factors that have a major function in sensory neurons, suggesting that they could act in a non-cell autonomous manner. Upon comparison with several lists of mRNAs shown to be deregulated 1, 5, or 7 days after nerve injury [41,42], we also observed, as suggested in [27], that 50% (26/52) of the transcriptional regulators deregulated upon *Adar1* deletion in the NC were associated with repair processes (Figure 1 and Appendix A). Finally, screening of the 52 differentially expressed mRNAs against the Interferome database [43] showed that 78.8% (41/52) are known to be regulated by type 1 or 2 IFN, and by downstream effectors of the Mda5/Mavs pathway (*Helz2*, *Irf1, −5, −7*, and *−9, Stat1*; “Interferome”; Figure 1 and Appendix A).

Based on these analyses, we chose to focus our experimental validation on the deregulated transcripts that (i) had a main function in NC and/or Schwann cells (in normal development or repair process), (ii) encoded proteins with complete or partial nuclear localization, and (iii) were listed within the Interferome database. We additionally retained *Tfap2b* based on the function of the *Tfap2a* and *Tfap2b* gene products in forming AP2 heterodimers during NC specification and differentiation [44], but excluded well-known Mda5/Mavs downstream effectors such as Interferon responsive factors (Irfs)- or Stat-encoding transcripts because their deregulation and putative function have already been validated in previous studies in other cell types [1,22,45].

Nine of the fifty-two transcriptional regulator transcripts were thus considered further: *Rxrg*, *Fosl2*, *Pml*, *Runx2*, *Egr1*, *Ddit3* (Chop), *Tfap2a*, *Tfap2b*, and *Fosl1* (indicated in red in Figure 1).

### 3.2. Egr1 Is Overexpressed and Tfap2a and Tfap2b Are Re-Activated in the Sciatic Nerves of Adar1cKO Mutants from Embryonic Day (E)18.5

Our previous study showed that *Adar1*cKO mutant mice presented Schwann-cell defects that occurred from E18.5 [27], in other words, the upregulation of ISG expression observed by RT-qPCR from this stage onwards, followed by a marked reduction in the expression of several transcripts encoding myelin markers (*Dusp15*, *Pmp22*, *Mpz*, and *Mbp*) from birth, along with a stall at the pro-myelinating stage observed by electronic microscope (EM) analyses.

We therefore investigated whether the expression of each of the selected transcription factor transcript is deregulated prior to, concomitantly with, or following these previously observed alterations by performing RT-qPCR on RNA extracted from the sciatic nerves of the controls and *Adar1*cKO mutants at two embryonic stages (E17.5 and 18.5) and at two post-natal time points (newborn and P4) (Figure 2). At E17.5, there was no difference between the *Adar1*cKO mutants and controls: three ISG (*Cxcl10*, *Isg15*, and *Rsad2*, constituting an ISG signature) and the nine transcription factors were expressed at similar levels in both. At E18.5, we observed strong upregulation of the ISG signature and a significant increase in *Egr1*, *Tfap2a*, and *Tfap2b* expression, but no modification in that of *Rxrg*, *Fosl2*, *Runx2*, *Ddit3*, or *Fosl1* in the *Adar1*cKO mutants relative to the control siblings. Although not significant, *Pml* expression increased twofold, but the upregulation of this transcript only became significant from post-natal day 0 (P0). Upregulation of *Fosl1* was also found from P0 onwards. Of note, the expression of *Rxrg*, *Fosl2*, *Runx2*, and *Ddit3* were all significantly upregulated only at post-natal day 4 in the *Adar1*cKO mice relative to the controls (Figure 2). Overall, these results suggest that the misexpression of *Egr1*, *Tfap2a*, and *Tfap2b* occurs from the earliest time points of *Adar1*cKO Schwann-cell anomalies and are maintained throughout all analyzed stages, whereas upregulation of the other transcription factors occurs secondarily. Although expressed at similar levels in the E13.5 dorsal root ganglia (DRG) of wild-type mice and *Adar1*cKO mutants, *Tfap2a* and *Tfap2b* were drastically downregulated from E17.5 onwards in the controls, but aberrantly re-expressed from E18.5 onwards in the *Adar1*cKO mutants (Appendix A). In contrast, *Egr1* expression was maintained at all stages analyzed in the controls, but strikingly upregulated from E18.5 onwards in the *Adar1*cKO mutants (Figure 2 and Appendix A).

We also tested whether the upregulation of these three transcription factors could be detected at the protein level by performing immunohistochemical staining on sciatic nerve sections of the *Adar1*cKO mutants and controls at P0 and P4. Unfortunately, no TFAP2B antibodies worked efficiently. Immunostaining of TFAP2A revealed a limited increase in small axons and no or faint expression in the Schwann cells of *Adar1*cKO mutants compared to the controls (Appendix A). Strikingly, the EGR1 protein showed a marked increase in nuclear staining and cell number in the *Adar1*cKO mutants compared to the controls (Figure 3). As shown in the quantifications, EGR1 was expressed in a limited percentage of cells (between 1.8% at birth and 6.2% at post-natal day 4) in the controls, but this percentage was increased by 10- and 5-fold in *Adar1*cKO compared to the controls at P0 and P4, respectively. The data in Figure 2 and Figure 3 thus suggest that *Adar1* deletion in the NC aberrantly increases both the *Egr1* mRNA and protein level.

### 3.3. Sciatic Nerves of Adar1; Mavs Double Mutants Show Unaltered Myelin and Normal Expression of Egr1, Tfap2a, and Tfap2b

We previously observed the partial rescue of myelin upon the extinction of *Ifih1* in the *Adar1*cKO Schwann cells in culture [27]. However, in vivo rescue by blocking the Mda5/Mavs signaling, along with the characterization of downstream effectors including transcription factors, remained to be investigated. To this end, we focused on the available *Adar^Δ2−13/Δ2−13^;Mavs^−/−^* double mutants [13] and analyzed the myelination process and expression of chosen transcription factors in these double mutants (hereafter referred to as *Adar1;Mavs* DM) compared to the controls. Of note, the deletion of *Adar1* in all tissues including the NC led to the premature death of *Adar^Δ2−13/Δ2−13^
*mutants at E11.5–E13.5, precluding the analysis of sciatic nerves in the animals of this genotype.

Because most of these double mutants died within the first 15 days of life (Appendix A), the sciatic nerves of *Adar1;Mavs* DM were analyzed by electron microscopy (EM) and compared to the controls at P1 (Figure 4A) and at P14 (surviving ones, Figure 4B). At the ultrastructural level, there was no observable difference between *Adar1;Mavs* DM and the other genotypes (referred to as controls) at both stages analyzed. Quantification confirmed these observations, showing an unaltered number of Schwann cells per square millimeter in *Adar1;Mavs* DM relative to the controls. Myelinated axons per square millimeter, the g-ratio (defined as the ratio of the inner axonal diameter to the total outer fiber diameter) and the number of Remak bundles per square millimeter were also not significantly different (Figure 4A,B). Of note, a general observation of these double mutants additionally revealed normal pigmentation (Appendix A).

To go further, transcriptomic analysis using RT-qPCR was additionally performed. It showed a similar expression of ISG (*Cxcl10*, *Isg15*, and *Rsad2*), late Schwann cell differentiation markers (*Pmp22*, *Mpz*, and *Mbp*), and transcription factors (*Egr1*, *Tfap2a*, and *Tfap2b*) in all sciatic nerves of *Adar1;Mavs* DM analyzed relative to the controls (Figure 4A,B).

Overall, these results show that *Adar1;Mavs* DM presented with normal myelination and unaltered expression of *Egr1*, *Tfap2a*, and *Tfap2b* relative to the controls in vivo as well as normal pigmentation. ADAR1 therefore safeguards Schwann cells from aberrant Mda5/Mavs-mediated ISG activation and the subsequent overexpression/reactivation of transcriptional regulators in vivo.

### 3.4. Deletion of One Copy of Egr1 Partially Rescues Adar1cKO Myelination Defects In Vivo, but Full Deletion Leads to the Premature Death of Mutant Mice

We next considered that if the observed *Adar1*cKO myelin defects were caused by overexpression/reactivation of one of these three early developmental regulators, then *Adar1*cKO phenotypic abnormalities should be rescued by their deletion in vivo. Based on the EGR1 overexpression observed at both the mRNA and protein level, we focused on the rescue potential of *Egr1* deletion.

Because mice (males and females) homozygous for the *Egr1* mutation are sterile [38,46], we used the crossing strategy shown in Figure 5A to generate *Adar1cKO* and *Egr1* single mutants, HtPA-Cre:*Adar1^fl/fl^;Egr1^LacZ/+^* (referred to as *Adar1*cKO;*Egr1*Het), HtPA-Cre:*Adar1^fl/fl^;Egr1^LacZ/LacZ^* double mutants (hereafter referred to as *Adar1*cKO;*Egr1* DM), and the controls. We first intended to analyze the phenotypes of mice at the peak of the peripheral myelination process (i.e., during the first two weeks after birth). To this end, we collected pups at the time of death or at P14, if still alive. We observed that the *Egr1* single mutants were smaller, but healthy and present at the expected Mendelian ratio (Figure 5A). *Adar1*cKO and *Adar1*cKO;*Egr1*Het pups died between P2 and P10 (Figure 5A and Appendix A) and were easily recognizable due to their almost total depigmentation (Appendix A), showing that the deletion of one copy of *Egr1* does not rescue the pigmentation defects nor the early lethality of *Adar1*cKO mutants.

We next dissected the sciatic nerves of animals of the mentioned genotypes at P4 to perform electron microscopy analysis (Figure 5B). Quantification revealed that the number of Schwann cells was similar in all genotypes (Figure 5B images and quantifications). The number of Remak bundles was significantly diminished in the *Egr1* single mutants, but was similar in all of the other genotypes. In addition, the Schwann cells established a 1:1 relationship with the axons and had begun wrapping a myelin membrane along the axons of the controls and *Egr1* mutants, whereas almost all of the *Adar1*cKO Schwann cells failed to initiate myelination. Strikingly, the Schwann cells had begun to wrap a myelin membrane along axons in four out of the five *Adar1*cKO;*Egr1*Het mice analyzed, suggesting that the deletion of one copy of *Egr1* partially rescues myelin formation in *Adar1*cKO mutants (Figure 5B).

The dissected sciatic nerves of animals of each genotype were also used to perform RT-qPCR to analyze the expression of (i) *Cxcl10*, *Isg15*, and *Rsad2*; and (ii) *Egr1*, *Tfap2a* and *Tfap2b* (Figure 5C). All of the *Adar1*cKO;*Egr1*Het mutants retained high levels of the expression of ISGs (similarly to *Adar1*cKO). As expected, the level of *Egr1* was drastically decreased in the *Egr1* mutants. It was also reduced in the *Adar1*cKO;*Egr1*Het mutants compared to *Adar1*cKO, but higher than in the controls. Although not significant, the expression of *Tfap2a* and *Tfap2b* was similarly upregulated in *Adar1*cKO;*Egr1*Het and *Adar1*cKO compared to the controls and *Egr1* single mutants. Altogether, these results suggest that the deletion of one copy of *Egr1* partially rescues the myelination process in *Adar1*cKO mutants. This partial rescue occurs despite ISG, *Tfap2a*, and *Tfap2b* expression remaining abnormally high in the *Adar1*cKO;*Egr1*Het mutants relative to the controls.

Strikingly, no *Adar1*cKO;*Egr1* DM pups were detected at birth (Figure 5A, dark orange), suggesting that complete *Egr1* deletion unexpectedly worsened the phenotype of the *Adar1*cKO mutants, precluding the analysis of myelin in the sciatic nerves of these double mutants in vivo.

### 3.5. E13.5 Adar1cKO;Egr1 DM Embryos Present with Craniofacial Alterations and Absence of Myelin Rescue In Vitro

Because myelination capacity can be assessed in vitro using mixed sensory neuron-Schwann cell co-culture starting from the dorsal root ganglia (DRG) from embryos at day E13.5, we collected eight litters of embryos at this stage. A total of 130 embryos were thus dissected, 15 of which were *Adar1*cKO;*Egr1* DM, suggesting that these embryos were present at the expected Mendelian ratio (percentage observed: 11.5%, expected: 6.25%). Of note, five of the *Adar1*cKO;*Egr1* DM embryos analyzed showed craniofacial alterations also found in pathologies related to neural crest defects such as mandibular hypoplasia, maxillomandibular fusion (syngnatia), and macroglossia (tongue abnormalities) (Figure 6A) [47,48,49].

We next tested the myelination capacity of three of the *Adar1*cKO;*Egr1* DM compared to the controls using the in vitro model above-mentioned. To this end, we first evaluated the relevance of this model by analyzing the expression of *Cxcl10*, *Isg15*, *Rsad2*, and *Egr1* by RT-qPCR at the beginning and end of the culture (when *Pmp22*, *Mpz*, and *Mbp* expression was detected and *Mbp*-positive segments were visible along the sensory neurons in control cultures but not those of *Adar1*cKO mutants [27]). Neither ISG nor *Egr1* expression were upregulated at early culture time points, but an upregulation of the three tested ISGs and *Egr1* were observed in the *Adar1*cKO mutant cultures relative to the controls at the end of the culture, highlighting the validity of this in vitro model (Figure 6B).

We therefore established similar cultures starting from *Adar1*cKO:*Egr1* DM, *Adar1*cKO;*Egr1*Het and compared them to those from *Adar1*cKO, *Egr1* single mutants and the controls. At the end of the 22 days of culture, we tested the presence of myelin segments labelled by Mbp along neurons labelled by Tuj1 by immunohistochemistry (Figure 6C). As previously published, we first observed that Mbp-positive segments (mark of myelination) were absent from the neurons (labelled by Tuj1) of *Adar1*cKO mutants relative to the controls [27]. In agreement with the partial rescue we observed in vivo, a significant number of Mbp-positive myelin segments were evidenced in two out of the three *Adar1*cKO:*Egr1*Het (quantification graph Figure 6C). However, none were observed in the 5 *Adar1*cKO;*Egr1* DM cultures, suggesting that the partial but not full deletion of *Egr1* in *Adar1*cKO mutants rescues myelination defects in vitro.

### 3.6. Overexpression of EGR1 in Schwann Cells Deregulates the Expression of Important EGR2 Target Myelin Genes

Among the four *Egr* (*Egr*1-4) family genes, EGR2 is the master regulator of myelination [35,36,50]. Mouse genetic experiments have shown that the induction of *Egr2* at the onset of myelination is critical: Schwann cells from *Egr2*-deficient mice fail to initiate myelin formation and EGR2 regulates the expression of a diverse array of genes required for peripheral nerve myelination and lipid/cholesterol synthesis [50,51,52]. In contrast, no deficiency in peripheral nerve myelination has been noted in two independent *Egr1* knockout lines including the one we used in this study [38,53].

Here, we speculated that the abnormal maintenance/overexpression of *Egr1* in pro-myelinating Schwann cells could affect EGR2 activity in *Adar1*cKO mutants. To explore this possibility, we scrutinized available ChIP-Seq datasets including (1) the ENCODE project consortium set that analyzed the binding pattern of 161 transcription factors across 91 cell types including EGR1 (full list annotated in [54,55]), and (2) a ChIP-Seq dataset from pooled sciatic nerves from rat pups at post-natal day 15, which describes EGR2 binding peaks, some of which were located within 100 kb of genes that are repressed or activated by EGR2, as determined by the microarray analysis of *Egr2* hypomorphic mice [52]. Cross comparison of these two datasets along with our own RNA-Seq data set revealed 174 genes in common (Figure 7A) and included genes mostly involved in myelin synthesis, lipid metabolism, and cholesterol synthesis (Appendix A). Unfortunately, no correlation between the binding identified by ChIP-Seq and the up- or down-regulation of EGR1 target genes is available. However, such information is available in the case of EGR2, prompting us to compare those to the ones obtained in the *Adar1*cKO mutant mice. Strikingly, 86% of the 174 genes of interest were downregulated in the *Adar1*cKO RNA-Seq and *Egr2* hypomorphic mutant, or vice versa (note that 40 EGR2-regulated genes also found on our list but not EGR1-regulated did not show this pattern) (Figure 7A and Appendix A). Despite the normal expression of *Egr2* in the sciatic nerves of *Adar1*cKO mutants [27], we therefore speculated that the overexpression of EGR1 might explain such observations.

To proceed further, we selected a short list of 10 genes among the 174 genes for in vitro experimental validation (*Mbp*, *Mpz*, *Hmgcr*, *Dhcr7*, *Dusp15*, *S100b*, *Atf3*, *Runx2*, *Lgals3*, *Pou3f1*, in Figure 7B). A mouse Schwann cell line (MSC, expressing basal level of EGR1 and EGR2 transcription factors and the chosen genes) was transfected, making use of an expression plasmid containing EGR1 cDNA. Forty-eight hours post-transfection, RT-qPCR experiments were performed to evaluate the expression levels of these selected genes in transfected cells compared to non-transfected MSC. Strikingly, *Mbp* and *Mpz, S100b, Atf3, Runx2* and *Lgals3* were significantly deregulated upon EGR1 overexpression in the same direction as the *Adar1*cKO mutants, suggesting that an inhibitory effect of EGR1 on six out of the ten selected EGR2-regulated myelin genes could contribute to the Schwann-cell defects observed in the *Adar1*cKO mutants.

## 4. Discussion

A cell-intrinsic mechanism for the initiation of autoinflammation/autoimmunity by immune stimulatory nucleic acids observed upon *ADAR1*/*Adar1* deletion or mutation has been described in patients and in various model organisms and tissues (for reviews, see [2,4,11,56,57]). A failure in A-to-I editing results in the recognition of unedited RNA by MDA5, recruitment of MAVS, inappropriate activation of type 1 IFN, ISG upregulation, and for most cell types, cell death [12,13,14,58]. Activation of protein kinase R (PKR, encoded by *Eif2ak2*), oligoadenylate synthetase (OAS)-RNase L pathways, and the roles of ZBP1 and LGP2 have also been recently reported [1,2,15,16,17,18,19,59]. Taken together, the interactions of ADAR1 with the MDA5, PKR, and ZBP1 dsRNA-sensing pathways demonstrate how ADAR1 utilizes its versatile domain architecture to safeguard the cell from dsRNA autoimmunity on multiple fronts. The A-to-I editing activity of the deaminase domain is responsible for preventing MDA5 multimerization; the dsRBD domain competes with PKR for dsRNA binding, and the Zα domain confers the unique ability to compete with ZBP1 for the binding of select dsRNAs [21,23,24]. Although *Mavs* deletion has been shown to rescue the embryonic lethality of *Adar1* mutants, *Pkr* deletion did not, and the deletion of *Zbp1* had a very limited effect. Nevertheless, triple deletions (*Adar1;Mavs;Zbp1* or *Adar1;Mavs;Pkr*) rescued embryonic and post-natal lethality, suggesting the synergic action of various pathways [15,18]. Once activated, the deregulation of some transcription factors and hundreds of ISG/ISR gene products has been shown, but their causality in the genesis of *Adar1* mutant phenotypes is yet to be determined. For example, double mouse mutant studies showed that deletion of the genes encoding the transcriptional regulators *Stat1* or *Irf3* did not rescue *Adar1* mutant defects, however, the deletion of *Irf7* in cardiomyocytes does (for reviews, see [1,22,45]). Aside from these well-known transcriptional effectors of the IFN pathway, the role of other transcription factors is yet to be determined.

Recently, we highlighted a role for *Adar1* in two NC derivatives: melanocytes and Schwann cells [27]. Because the coordinated expression of several transcription factors is known to increase, in particular, between the pro-myelinating and myelinating stage (the stage affected in *Adar1*cKO mutants), whereas others are known to decrease strongly [32,35,60,61,62], we speculated that the deregulation of some transcriptional regulator could play a role in the genesis of the *Adar1*cKO mutant peripheral nervous system phenotype. We therefore re-evaluated our previously published RNA-Seq dataset along with a new one and thus identified 52 transcriptional regulators that were up- or downregulated in the *Adar1*cKO mutants relative to the controls with an FC ≥ 5. The RT-qPCR and immunohistochemistry experiments showed that the overexpression of *Egr1* (both at the mRNA and protein level) and reactivation of *Tfap2a* and *Tfap2b* occurred from E18.5 (i.e., concomitantly with the observed Schwann cell alterations), whereas deregulation of the other six transcription factors selected for experimental validation occurred secondarily. Of interest, the upregulation of these three transcription factors was found to be concomitant with ISG upregulation, and their expression was unaltered in the sciatic nerves of *Adar1;Mavs* DM pups relative to the controls, suggesting that although we cannot exclude the involvement of other pathways such as PKR, the recognition of unedited RNAs by Mda5, Mavs recruitment, and IFN/ISG activation contribute to the aberrant maintenance/re-activation of *Egr*1, *Tfap2a*, and *Tfap2b* in vivo.

Strikingly, the deletion of one copy of *Egr1* partially rescued the myelin defects observed in the *Adar1*cKO mutants. Indeed, electron microscopy showed that a significant number of Schwann cells had begun to wrap a myelin membrane along axons in the sciatic nerves of four out of the five *Adar1*cKO;*Egr1*Het pups analyzed, an event that was never observed in the *Adar1*cKO mutants ([27] and this study). The expression of ISGs, *Tfap2a*, and *Tfap2b* (although not significant for the last 2) remained aberrantly upregulated relative to the controls. In vitro, most *Adar1*cKO;*Egr1*Het also presented with the partial rescue of myelination defects compared to *Adar1*cKO. Overall, these results suggest that the deletion of one copy of *Egr1* partially rescues the myelination defects associated with *Adar1* deletion in NC cells, and suggest that EGR1 mainly acts downstream of Mda5/Mavs pathway activation, but independently/parallel to TFAP2A and TFAP2B (see model in Figure 8).

This partial rescue appears, however, to be tissue specific. Indeed, the *Adar1*cKO and *Adar1*cKO;*Egr1*Het animals all died between P2 and P10 and showed complete depigmentation, demonstrating no survival increase and no rescue of skin melanocytes upon the deletion of one copy of *Egr1*.

Strikingly, no *Adar1*cKO;*Egr1* DM pups was detected at birth, suggesting that *Egr1* knockout unexpectedly worsened the phenotype of the *Adar1*cKO mutants. Of note, the expected number of *Adar1*cKO;*Egr1* DM embryos were observed at E13.5, but a significant number of them presented craniofacial defects, suggesting additional alterations. Additionally, mixed cultures of embryonic DRGs showed no rescue of myelination defects in vitro, suggesting that too much or not enough EGR1 is detrimental in the context of *Adar1* NC-specific knockout. EGR1 is expressed in nearly all organs and cell types, and beyond the NC model studied here, we suggest that the role of EGR1 in other tissue/cell types affected upon *Adar1* deletion in other models should be investigated.

Among the four *Egr* (*Egr*1-4) family genes, EGR2 is the master regulator of myelination [35,36,50]. Mouse genetic experiments have shown that the induction of *Egr2* at the onset of myelination is critical: Schwann cells from *Egr2*-deficient mice fail to initiate myelin formation, and Egr2 regulates the expression of a diverse array of genes required for peripheral nerve myelination and lipid/cholesterol synthesis [50,51,52]. Of note, the mutually exclusive expression of *Egr1* and *Egr2* has been shown to be crucial for the appropriate differentiation of various cell types [46,63]. Here, we show that the overexpression of *Egr1* in *Egr2* expressing cells alters the myelination process in vivo, and that the overexpression of EGR1 inhibits or activates the expression of a few EGR2 target genes in a mirrored pattern, suggesting that the abnormal maintenance/overexpression of EGR1 in pro-myelinating Schwann cells could affect EGR2 activity in *Adar1*cKO mutants.

Other functions of EGR1, such as regulation of the stress response, might also play a role [64,65,66]. Indeed, this gene is rapidly expressed after a number of stimuli such as oxygen deprivation or exposure to growth factors, cytokines, stress, or injury. Both proviral (enhanced VEEV, MHV, and EV71 replication) and antiviral (suppressed FMDV and Seneca Valley virus replication) actions of EGR1 have been described [66]. Of interest, upon FMDV infection, EGR1 upregulation has been shown to enhance TBK1 phosphorylation, promoting the activation of type I IFN signaling and the subsequent suppression of viral replication [66]. The overexpression of *Egr1* was also shown to significantly promote poly (I:C)-induced type I IFN signaling, suggesting a positive regulatory role of this transcription factor on type 1 IFN signaling [66]. In our study, the expression of *Egr1* was normal in rescued *Adar1;Mavs* DM, and ISG expression remained high in the *Adar1*cKO;*Egr1*Het mice, suggesting that EGR1 acts downstream of Mda5/Mavs, but a feedback loop cannot be excluded.

Unexpectedly, the total deletion of *Egr1* exacerbates the phenotype of *Adar1*cKO animals. Too much or not enough EGR1 thus seems deleterious for the development of NC-derived structures, but the underlying molecular mechanism remains to be determined. Interestingly, a recent publication has suggested that the association between EGR1 and inflammation might be more complex than expected. Indeed, EGR1 has been ascribed a role as a gatekeeper of inflammatory enhancers in human macrophages [67]. In this cell type, EGR1 associates with the nucleosome remodeling and deacetylation (NuRD) chromatin remodeler and represses hundreds of enhancers of pro-inflammatory genes, blunting their activation and the immune response. Whether similar mechanisms are at work in some neural crest-derived cells (and at the origin of *Adar1*cKO;*Egr1* DM lethality) as well as in other tissues should be investigated in the future.

Of note, *Tfap2a* and *Tfap2b* expression remained high in *Adar*1cKO;*Egr1*Het mutants, an observation that led us to speculate that they could also participate only in the partial rescue observed in *Adar*1cKO;*Egr1*Het. Previous publications have suggested that the loss of *Tfap2a* is insufficient to prevent the generation of Schwann-cell precursors or Schwann cells (no defects at E13 or E17), but its overexpression reduces the number of Schwann cells in culture [68]. TFAP2a misexpression was also shown to contribute to the genesis of the phenotype of *Ep400* mutant mice [69]. The alteration of late Schwann-cell development in *Ep400* mutants was proposed to result from a failure to shut off early developmental regulators whose continued presence in differentiating Schwann cells interferes with the maturation and myelination process [69]. The transcription factors TFAP2A, SOX2, POU3F3, PAX3, SOX1, and SOX3 were strongly upregulated in the post-natal sciatic nerves of these mutant mice. The authors showed that Schwann-cell specific-*Ep400;Tfap2a* double knockout mice were less strongly affected than the *Ep400* single mutants, suggesting that concomitant *Tfap2a* deletion leads to a substantial, although incomplete, rescue of most of the defects caused by the absence of *Ep400* [69]. Similarly, *Tfap2a* misexpression (along with that of *Tfap2b*, with which it interacts) could contribute to the phenotype of the *Adar1*cKO mutants described in this study. Although Tfap2a seems to be mainly expressed in sensory neurons (immunohistochemistry experiments), the Schwann cell-specific deletion of *Adar1* and *Tfap2a* should help test this possibility in the future.

As previously mentioned by others, the levels of transcriptional regulators of the immature state have been found to be elevated in patients with Charcot–Marie–Tooth disease and mouse models of this peripheral neuropathy [70,71,72]. Although the consequences of such an increase are complex, they highlight the relevance of our findings in the context of human disease.

## 5. Conclusions

Our study highlights the contribution of a small number of transcription factors to the defects observed upon *Adar1* deletion in the NC, and possibly links A-to-I post-transcriptional and transcriptional regulation to the myelination process of the peripheral nervous system. In particular, our results argue that the absence of *Adar1* in NC derivatives results in the failure to shut-off early developmental regulators among which is EGR1, the continued presence of which in differentiating Schwann cells interferes with the maturation and myelination process.

## Figures and Tables

**Figure 1 cells-13-01952-f001:**
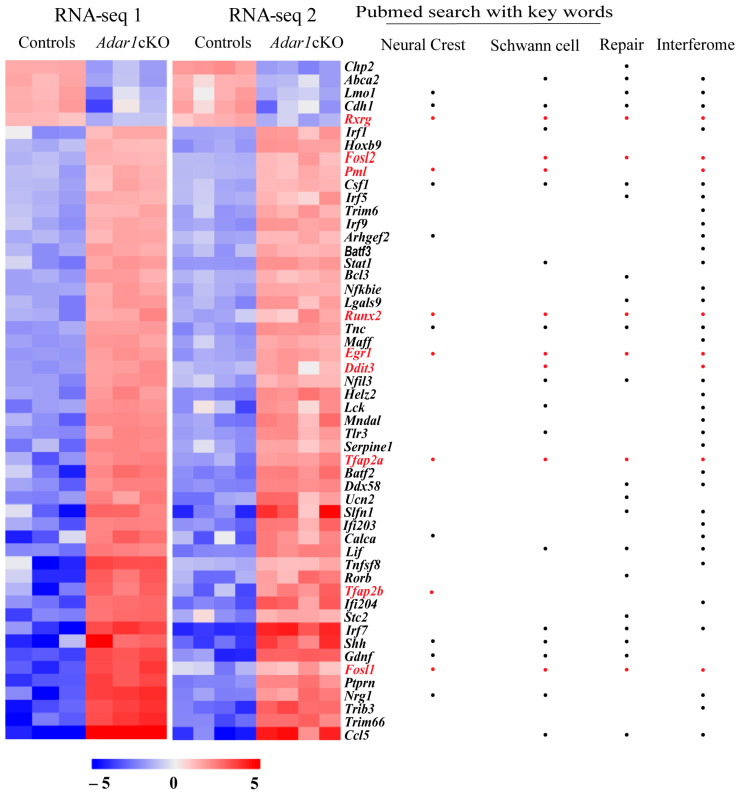
Transcriptional regulator transcripts deregulated in the sciatic nerves of *Adar1*cKO versus the controls. Transcriptomic analysis identified 52 transcriptional regulators deregulated in the sciatic nerves of HtPA-Cre; *Adar1^fl/fl^* (Adar1cKO) mutants relative to the controls with fold change (FC ≥ 5). Heatmap representations of differentially-expressed transcriptional regulators [found in association with the GO terms mentioned in the text and deregulated more than fivefold in *Adar1*cKO mutants (n = 7) relative to the controls (n = 7) in RNA-Seq 1 and RNA-Seq 2 datasets]. The name of each gene is indicated on the right, with whether it belongs to the Interferome database, or involved in the Schwann cell repair process (see Appendix A for PMIDs). Genes found in the literature using the “neural crest” or “Schwann cells” keywords are also indicated. Colors reflect the z-score. Deregulated transcripts indicated in red (including dots) were the ones selected for further experimental validation.

**Figure 2 cells-13-01952-f002:**
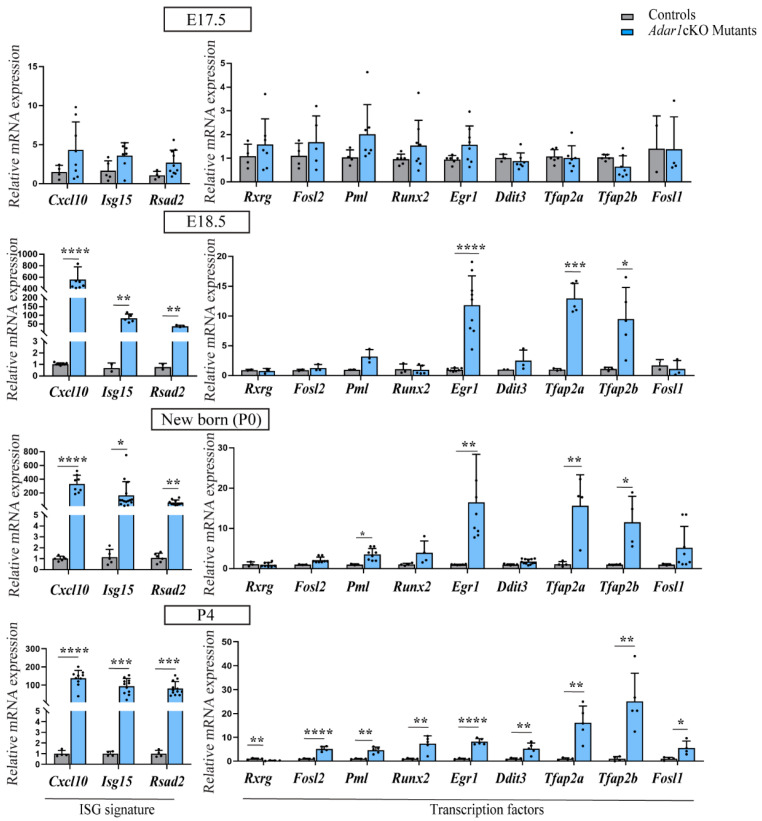
Timing of the expression of candidate transcription factors in the sciatic nerves of *Adar1*cKO versus the controls. Analysis of the expression of *Rxrg*, *Fosl2*, *Pml*, *Runx2*, *Egr1*, *Ddit3*, *Tfap2a*, *Tfap2b*, *Fosl1*, and Interferon stimulated genes (ISG signature composed of *Cxcl10, Isg15, Rsad2*) in the sciatic nerves of the controls (grey) and HtPA-Cre; *Adar1^fl/fl^* (*Adar1*cKO) mutants (blue) at embryonic day E17.5 and E18.5, newborn (P0), and post-natal day 4 (P4) by RT-qPCR. All data represent the mean ± SD. Statistical differences between the groups (n = 3 to 12 controls and n = 3 to 11 mutants) were determined using the *t*-test (asterisks represent *p* values: * *p* ≤ 0.05, ** *p* < 0.01, *** *p* < 0.001, **** *p* < 0.0001).

**Figure 3 cells-13-01952-f003:**
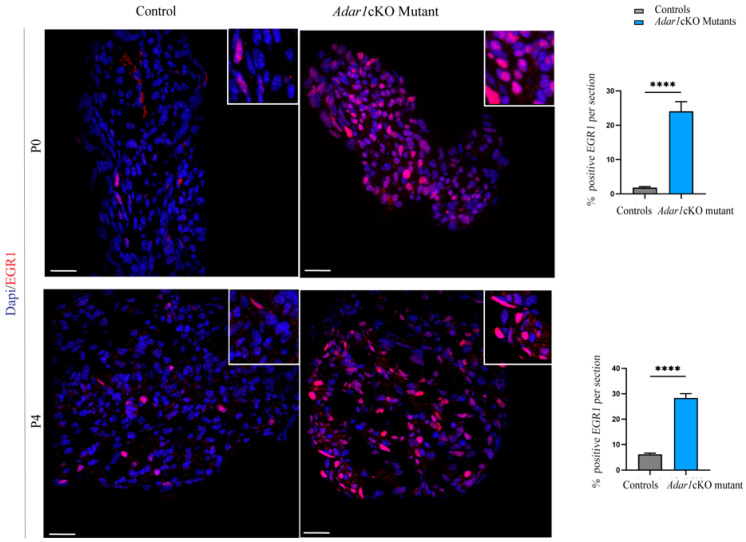
Upregulation of the EGR1 protein in the sciatic nerves of *Adar1*cKO compared to the controls. EGR1 (red) immunostaining was performed on sections of sciatic nerves of *Adar1*cKO and the control mice at P0 and P4. Counter staining with DAPI is shown to identify the nuclei. Scale bar: 60 µm. Quantification was performed by counting the percentage of EGR1 positive nuclei over the total number of DAPI positive cells per section of 9 to 12 sections per genotype. Statistical differences between the groups were determined using the *t*-test (asterisks represent *p* values: **** *p* < 0.0001).

**Figure 4 cells-13-01952-f004:**
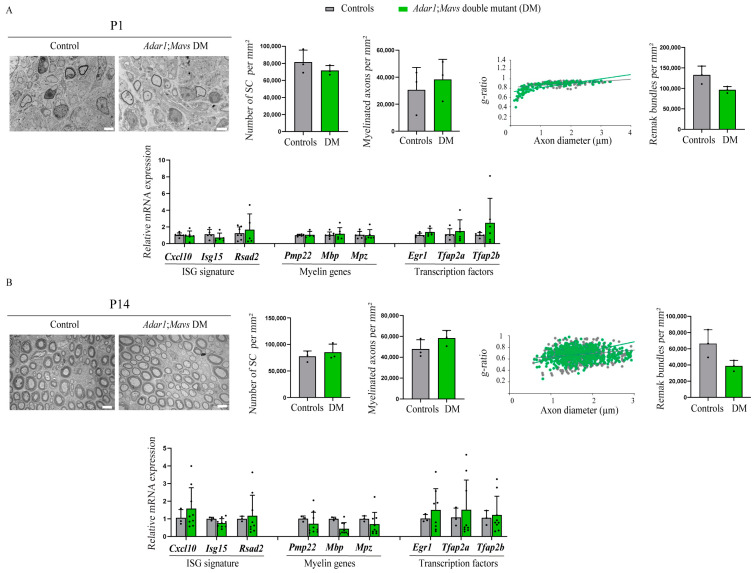
Axon myelination and normalized expression of *Egr1*, *Tfap2a*, and *Tfap2b* in the sciatic nerves of *Adar1;Mavs* DM versus the controls. Electron micrographs of transverse sections of sciatic nerves from *Adar^Δ2−13/Δ2−13^;Mavs^−/−^* double mutant (*Adar1;Mavs* DM) mice at P1 (**A**) and P14 (**B**), showing the presence of myelin in double mutants compared to the controls (*Adar**^Δ2−13/+^**; Mavs ^−/−^*, or *Adar1^+/+^; Mavs^−/−^*). Quantifications, presented as the mean ± SD, were performed on at least 5 EM images (means presented) from n = 3 controls and n = 3 mutants at each stage to count the Schwann cell number, myelinated axon number, Remak bundles per square millimeter, and g-ratio in the controls (grey) and *Adar1;Mavs* DM (green), as described in [40]. Statistical differences between the groups were determined using the *t*-test. Scale bar: 5 µm. The lower halves of panels (**A**,**B**) show the expression of the ISG (ISG signature composed of *Cxcl10*, *Isg15*, *Rsad2*) of transcripts encoding myelination proteins (*Pmp22*, *Mbp*, and *Mpz*), and of *Egr1*, *Tfap2a*, and *Tfap2b* in the sciatic nerves of the controls (grey) and *Adar1;Mavs* DM (green) at indicated stages. Statistical analyses were made between the groups (n = 4 controls and n = 7 DM at P1 and n = 3 controls and n = 3 DM at P14). Statistical differences between the groups were determined using the *t*-test.

**Figure 5 cells-13-01952-f005:**
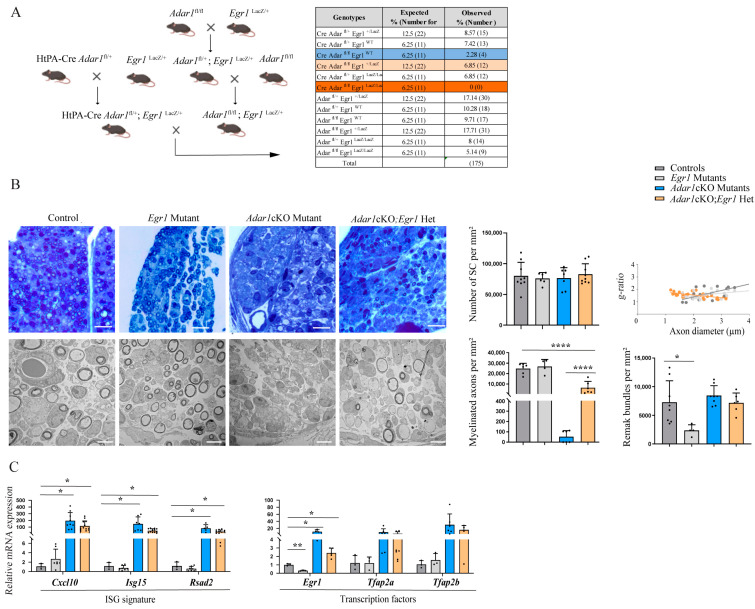
Deletion of one copy of Egr1 rescues *Adar1*cKO myelination defects in vivo. (**A**) Breeding strategy used to generate HtPA-Cre; *Adar1^fl/fl^; Egr1^LacZ/LacZ^* (*Adar1cKO;Egr1* DM), HtPA-Cre; *Adar1^fl/fl^; Egr1^LacZ/+^* (*Adar1cKO;Egr1*Het), single mutants, and the controls. The percentage (%) and expected number of animals of each genotype versus those collected between P1 and P14 are presented in the table. Comparison to the expected number was performed using the chi-square test to measure the significance of the deviation from Mendelian expectations (and to estimate the percentage survival of each genotype). This test revealed that the two sets of data, the observed and expected values, were different *p* = 0.00006711. (**B**) Semi-thin and electron micrographs of transverse sections of sciatic nerves from the controls (grey), *Egr1* (light grey) and *Adar1*cKO (blue) single mutants, and HtPA-Cre; *Adar1^fl/fl^; Egr1^LacZ/+^* (*Adar1*cKO;*Egr1*Het, orange) at P4. Quantification was performed by counting the number of Schwann cells, myelinated axons, and Remak bundles per square millimeter on two to six pictures of n = 3 animals per group and the g-ratio as described in [40]. Statistical differences between the groups were determined using the *t*-test (Asterisks represent *p* values: * *p* ≤ 0.05, **** *p* < 0.0001). Scale bar: 5 µm (EM) and 0.7 µm (semi-thin). (**C**) RT-qPCR, represented as the mean  ±  SD, were performed to quantify (i) *Cxcl10*, *Isg15* and *Rsad2*, (ii) *Egr1*, *Tfap2a*, and*Tfap2b* in the sciatic nerves of the controls (grey), *Egr1* single mutants (light grey), *Adar1*cKO (blue), and HtPA-Cre; *Adar1^fl/f^;Egr1^LacZ/+^* (*Adar1*cKO;*Egr1*Het, orange) relative to the controls. Note the partial myelin rescue in *Adar1*cKO;*Egr1*Het compared to *Adar1*cKO despite ISG signature activation and *Tfap2a* and *Tfap2b* overexpression. Statistical differences between the groups were determined using the *t*-test (asterisks represent *p* values: * *p* ≤ 0.05, ** *p* < 0.01).

**Figure 6 cells-13-01952-f006:**
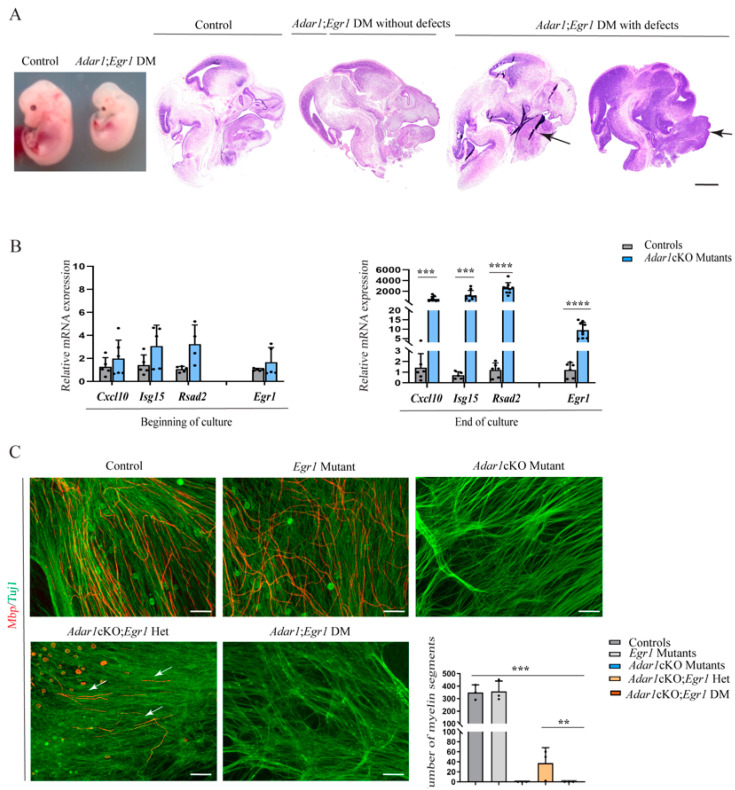
Craniofacial alterations and myelin defects in *Adar1*cKO;*Egr1* DM E13.5 embryos. (**A**) photo of whole E13.5 *Adar1*cKO;*Egr1* DM embryos versus sibling controls, and sagittal sections of E13.5 *Adar1*cKO;*Egr1* DM embryos and controls stained with H&E showing examples of craniofacial alterations observed in 5 out of the 15 *Adar1*cKO;*Egr1* DM (macroglossia and syngnatia are indicated with black arrows). Scale bar: 1 mm. (**B**) Relative expression of *Egr1* and three ISG in the mRNA extracted from primary cultures of mixed neurons and Schwann cells from dorsal root ganglia (DRG) of the controls (grey) and *Adar1*cKO E13.5 embryos (blue) at the start and end of the culture. Statistical differences between the groups (n = 3 to 8) were determined using the *t*-test (asterisks represent *p* values: *** *p*  <  0.001, **** *p*  <  0.0001). (**C**) End-stage mixed embryonic DRG cultures established from the controls, *Egr1* or *Adar1*cKO single mutants, *Adar1*cKO;*Egr1*Het and *Adar1*cKO;*Egr1* DM embryos immunostained with Mbp and Tuj1 markers to visualize myelin segments (red) along sensory axons (green) at the end of the culture. Scale bar: 24 µm. Graph shows the quantification of the number of myelin segments per axons per well in n = 3 cultures established from the embryos of the indicated genotypes. Statistical differences between the groups were determined using the *t*-test (asterisks represent *p* values: ** *p* < 0.01 and *** *p* < 0.001).

**Figure 7 cells-13-01952-f007:**
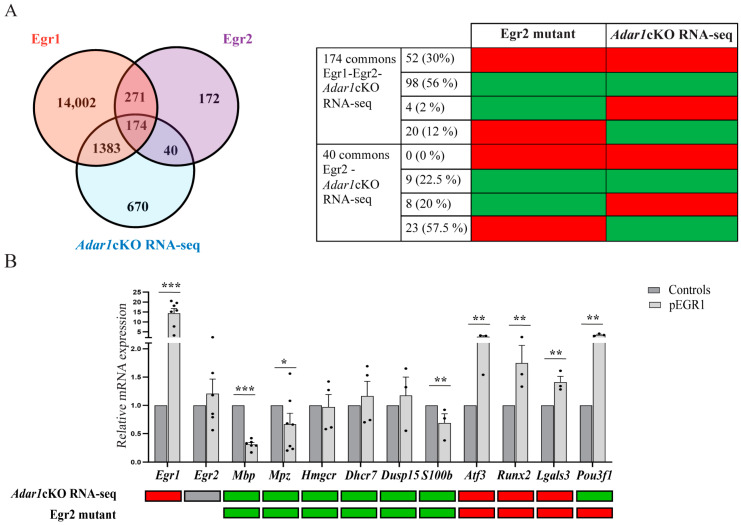
Overexpression of EGR1 disturbs the expression of several EGR2 regulated genes. (**A**) Venn diagram showing the overlap between genes misregulated in the RNA-Seq data generated in this study (*Adar1*cKO RNA-Seq), in EGR2-ChIP-Seq data on sciatic nerves [52], and in EGR1-ChIP-Seq performed in various cell lines [55]. The number of genes overlapping between each dataset are indicated. Table on the right indicates the number and % of genes upregulated (red) or downregulated (green) in the Egr2 hypomorphic mutants or in *Adar1*cKO mutants among the 174 genes and 40 genes overlapping between EGR1/EGR2 ChIP-Seq/*Adar*1cKO RNA-Seq or EGR2/*Adar1*cKO datasets, respectively. (**B**) RT-qPCR validation of 10 of the overlapping and differentially-regulated genes on RNA extracted from mouse Schwann cells non-transfected or transfected with the EGR1 overexpression plasmid. Relative expression levels in non-transfected (grey) and transfected (light grey) are presented as the mean  ± SEM Statistical differences between the groups (n = 3 to 7) were determined using the *t*-test (asterisks represent *p* values: * *p* ≤ 0.05, ** *p* < 0.01, *** *p* < 0.001. For comparison, up (red) and down (green) expression of selected transcripts in the *Adar1*cKO RNA-Seq and EGR2 mutant datasets are presented below.

**Figure 8 cells-13-01952-f008:**
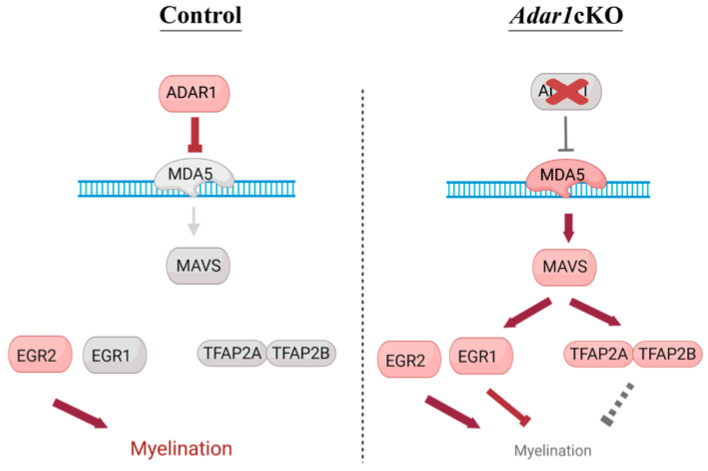
Model summarizing the proposed effect of *Adar1* deletion, leading to increased expression of the transcription factor. In normal Schwann cells, ADAR1 edits dsRNAs, preventing their recognition by the MDA5 sensor. Upon NC-specific *Adar1* deletion, activation of the MDA5/MAVS pathway leads to aberrant overexpression/re-expression of *Egr1*, *Tfap2a*, and *Tfap2b*, which are repressors of the myelination process; EGR1 competes with EGR2 to control the myelination process. Red: activated and grey: not activated.

## Data Availability

The data that support the findings of this study are available within the text and Appendix A. The previously published RNA-Seq 1 data are available under GSE127795 and the new RNA-Seq 2 data have been submitted to the Annotare repository under the accession number ArrayExpress E-MTAB-11197.

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
