# Peer review of "Overexpression of Egr1 Transcription Regulator Contributes to Schwann Cell Differentiation Defects in Neural Crest-Specific Adar1 Knockout Mice"

_cells, 2024, doi:10.3390/cells13231952_

Round 1
Reviewer 1 Report
Comments and Suggestions for Authors
The authors did extensive investigation of the molecular players linking Adar1 and Schwann cell mediated myelination using multiple approaches including transgenic animals and classic genetics, RNA-Seq and bioinformatic analysis, qPCR, immunostaining, electron microscopy and plasmid-based overexpression assay. Although the results were not fully in line with their original working hypothesis, the authors did frankly discuss about the limits, alternative explanations and future directions of the study. This is understandable as biological systems are complex. Gene regulatory network is filled with different kinds of feedback and feedforward loops, rather than a straight line. However, there are still some major issues the authors need to address before publication, such as the missing Figure 6, redundant paragraphs on Page 9, inverse changes of Pou3f1 expression levels in Egr1 overexpressed mouse Schwann cell line and Adar1cKO sciatic nerves.
Sentence-by-sentence comments:
Page 2, line 50: Please change “DSH1” to “DSH”, as DSH is a disease, not a gene.
Page 2, line 79: Please add “Being” before “multipotent”.
Page 2, line 82: “specific mesenchymal populations within nerves”? That does not sound right. Maybe the authors mean “specific mesenchymal populations at cranial locations”, like the chondrocytes forming the mandible?
Page 2, line 90: Please add “of Schwann cell development” after “stalling”.
Page 9, line 339-342: This part is exactly the same as line 353-356.
Page 14, Figure 6 is missing.
Page 15, line 578: Pou3f1 expression is upregulated upon Egr1 overexpression in the mouse Schwann cell line, whereas its expression is downregulated in Adar1cKO sciatic nerves. This cannot be described as “in the same direction”.
Page 16, line 599: What do the authors mean by “ADAR1/Adar1”? One refers to the protein and the other refers to the gene? Then why both are in italics?
Page 19, line 705: Similarly, do the authors refer to both the gene and the protein by using the term “Egr1/EGR1”?
Author Response
Response to reviewer 1 comments
Thank you very much for taking the time to review this manuscript. Please find the detailed responses below and the corresponding revisions/corrections in track changes in the re-submitted files
Point-by point response to comments and suggestions
The authors did extensive investigation of the molecular players linking Adar1 and Schwann cell mediated myelination using multiple approaches including transgenic animals and classic genetics, RNA-Seq and bioinformatic analysis, qPCR, immunostaining, electron microscopy and plasmid-based overexpression assay. Although the results were not fully in line with their original working hypothesis, the authors did frankly discuss about the limits, alternative explanations and future directions of the study. This is understandable as biological systems are complex. Gene regulatory network is filled with different kinds of feedback and feedforward loops, rather than a straight line. However, there are still some major issues the authors need to address before publication, such as the missing Figure 6, redundant paragraphs on Page 9, inverse changes of Pou3f1 expression levels in Egr1 overexpressed mouse Schwann cell line and Adar1cKO sciatic nerves.
Comment 1. Page 2, line 50: Please change “DSH1” to “DSH”, as DSH is a disease, not a gene.
We agree with this comment. Page 2 line 50 you can now read “DSH”
Comment 2. Page 2, line 79: Please add “Being” before “multipotent”.
Agree. Page 2 line 79 you can now read “Being multipotent, these cells can detach…”
Comment 3. Page 2, line 82: “specific mesenchymal populations within nerves”? That does not sound right. Maybe the authors mean “specific mesenchymal populations at cranial locations”, like the chondrocytes forming the mandible?
Thank you for pointing this out. Page 2 line 80-81 you can now read “…chromaffin cells of the adrenal medulla and specific mesenchymal populations at cranial locations”
Comment 4. Page 2, line 90: Please add “of Schwann cell development” after “stalling”.
Agree. We have accordingly revised the text page 2 line 90 “stalling of Schwann cell development at the pro-myelinating stage”
Comment 5. Page 9, line 339-342: This part is exactly the same as line 353-356.
Thank you for pointing this out. Page 9 lines 339-344 have been removed.
Comment 6. Page 14, Figure 6 is missing.
The Editorial Office layout missed Figure 6, which has been added.
Comment 7. Page 15, line 578: Pou3f1 expression is upregulated upon Egr1 overexpression in the mouse Schwann cell line, whereas its expression is downregulated in Adar1cKO sciatic nerves. This cannot be described as “in the same direction”.
Thank you for pointing this out. Page 16 lines 586-589 you can now read “Strikingly, Mbp and Mpz, S100b, Atf3, Runx2 and Lgals3 are significantly deregulated upon EGR1 overexpression in the same direction as in Adar1cKO mutants, suggesting that an inhibitory effect of EGR1 on 6 out of the 10 selected EGR2- regulated myelin genes could contribute to the Schwann cell defects observed in Adar1cKO mutants.”
Comment 8. Page 16, line 599: What do the authors mean by “ADAR1/Adar1”? One refers to the protein and the other refers to the gene? Then why both are in italics?
ADAR1/Adar1 refers to Human and mouse genes, both in italics.
Comment 9. Page 19, line 705: Similarly, do the authors refer to both the gene and the protein by using the term “Egr1/EGR1”?
Egr1/EGR1 refers to mouse and Human genes, both in italics. To avoid misunderstanding, the sentence now page 20 line 717 has been modified as follow “Indeed, this gene is rapidly expressed….”

Reviewer 2 Report
Comments and Suggestions for Authors
Zerad et al. have submitted an attractive review manuscript on the role of EGR1 and ADAR1 in Schwann cells differentiation.
Considering the quality of the research and the importance of the topic, I suggest its acceptance for publication after minor revision.
Minor comments:
1-Overall, although the quality of the research work is good, as well as the English, the whole text is too dense while reading, which often makes it a bit difficult to understand and to follow. I definitely recommend to summarize and rewrite the manuscript, particularly parts in which concepts, methodology or previous published results are repeated and strenuously described, in order to make the text flow and be more understandable. For instance: please split long paragraphs into smaller one: specially lines 428-467; 544-581; 704-726 and so on.
2-Importantly, authors are full accountable to ensure that they are not presenting here any data or figures already published in their previous work (ref. 27).
3-Methods: Section 2.4 Please, describe the set up for the PCR cycles and Tm used.
4-3. Results: Please, avoid retelling methodology in the results section or in the Figure legends.
Lines 255-256, please add the reference to the published data that you are referring to!
Please, correct the format of page 9: Table legend and the text paragraph are repeated.
Figure 6 is missing (!!)
5-4. Discussion: It is definitely too long. Please, do not repeat results and methodology in the discussion section, for instance, lines 641-655, lines 659-660 are a repetition the results section (!). Keep to discuss the results and argument prospects.
Paragraph from lines 622-639 is redundant, not necessary and too long. If you need to cite here your previous work, please do it in just 1-2 lines.
6-5. Conclusion: Please, correct the capital letters.
The conclusion should be shorter and re-focused to give a concise “take home“-message.
Please, remove the cites and references of this section (!!). Lines 751-754 should be moved to the discussion section.

Author Response
Response to reviewer 2 comments
Thank you very much for taking the time to review this manuscript. Please find the detailed responses below and the corresponding revisions/corrections in track changes in the re-submitted files
Point-by point response to comments and suggestions
Zerad et al. have submitted an attractive review manuscript on the role of EGR1 and ADAR1 in Schwann cells differentiation. Considering the quality of the research and the importance of the topic, I suggest its acceptance for publication after minor revision.
Comment 1. -Overall, although the quality of the research work is good, as well as the English, the whole text is too dense while reading, which often makes it a bit difficult to understand and to follow. I definitely recommend to summarize and rewrite the manuscript, particularly parts in which concepts, methodology or previous published results are repeated and strenuously described, in order to make the text flow and be more understandable. For instance: please split long paragraphs into smaller one: specially lines 428-467; 544-581; 704-726 and so on.
- Long paragraph previously lines 428-467 is now presented in 3 paragraphs lines 430-449, 450-462 and 463-474. The first paragraph was rewritten to remove informations already presented in methods section (see comment 4). You can now read “Because mice (males and females) homozygous for the Egr1 mutation are sterile (38, 49), we used the crossing strategy shown in Figure 5A to generate Adar1cKO and Egr1 single mutants, HtPA-Cre:Adar1fl/fl;Egr1LacZ/+ (referred to as Adar1cKO;Egr1Het), HtPA-Cre:Adar1fl/fl;Egr1LacZ/LacZ double mutants (hereafter referred to as Adar1cKO;Egr1 DM), and controls. We first intended to analyze the phenotypes of mice at the peak of the peripheral myelination process, i.e., during the first two weeks after birth. To this end, we collected pups at the time of death or at P14, if still alive. We observed that the Egr1 single mutants were smaller, but healthy and present at the expected Mendelian ratio (Figure 5A). Adar1cKO and Adar1cKO;Egr1Het pups died between P2 and P10 (Figure 5A and Figure S4A) and were easily recognizable due to their almost total depigmentation (Figure S4B and C), showing that deletion of one copy of Egr1 does not rescue the pigmentation defects nor the early lethality of Adar1cKO mutants.” The second paragraph was also slightly modified as follows “We next dissected the sciatic nerves of animals of mentioned genotypes at P4 to perform electron microscopy analysis (Figure 5B). Quantifications revealed the number of Schwann cells was similar in all genotypes (Figure 5B images and quantifications). Number of remark bundles was significantly diminished in Egr1 single mutants, but it was similar in all the other genotypes. In addition, Schwann cells had established a 1:1 relationship with the axons and had begun wrapping a myelin membrane along the axons of controls and Egr1 mutants, whereas almost all the Adar1cKO Schwann cells failed to initiate myelination….”
- Long paragraph previously lines 544-581 is now presented in 3 paragraphs lines 552-558 + 559-579+ 580-590. We slightly shortened the section lines 559-579 by removing unnecessary information including “among which are 44985 events corresponding to an EGR1 binding in 14002 different genes” line 563, “Sprague-Dawley” line 565, “7544 peaks corresponding to” line 565, “(Adar1cKO RNA-seq)”line 569 and “in the ENCODE project consortium” line 572.
- Long paragraph previously lines 704-726 is now presented in 2 paragraphs lines 716-728 and 729-739.
Comment 2. Importantly, authors are full accountable to ensure that they are not presenting here any data or figures already published in their previous work (ref. 27).
Thank you for pointing this out. We confirm that we are not presenting data or figures already presented in ref 27. All figures have been generated from unpublished experiments.
Comment 3. Methods: Section 2.4 Please, describe the set up for the PCR cycles and Tm used.
The information has been added page 4 lines 184-185. You can now read “The PCR settings were 95°C for 10 min, then 40 cycles of denaturation at 95°C for 15 sec, followed by annealing and extension at 60°C for 1 min.”
Comment 4. 4.3. Results: Please, avoid retelling methodology in the results section or in the Figure legends.
To answer this comment, we removed the sentence “The relative abundance values of each amplicon was normalized to the internal control β-actin, and expression level of each amplicon in mutants and controls is presented relative to controls.” In figure 2, figure 4 and figure 5 legends (lines 350-52, 418-19 and 497-99 respectively). The note at the end of figure 4 was also removed (line 422). We also removed redundant explanation of g ratio in legends of figure 5 (491-92). We finally removed the description of the crosses that were already depicted in methods and in Figure 5A (see answer to comment 1, lines 430-35 and 443-45).
Comment 5. lines 255-256, please add the reference to the published data that you are referring to!
Agree. Reference 27 is now indicated.
Comment 6. Please, correct the format of page 9: Table legend and the text paragraph are repeated.
Thank you for pointing this out. Page 9 lines 339-344 have been removed.
Comment 7. Figure 6 is missing (!!)
The Editorial Office layout missed Figure 6, which has been added.
Comment 8. Discussion: It is definitely too long. Please, do not repeat results and methodology in the discussion section, for instance, lines 641-655, lines 659-660 are a repetition the results section (!). Keep to discuss the results and argument prospects.
- Section 641-655 is now shortened. To this end, we have rephrased the first part lines 652-55. You can now read “We therefore revaluated our previously published RNA-seq dataset along with a new one and thus identified 52 transcriptional regulators that were up- or down-regulated in the Adar1cKO mutants relative to the controls with a FC ≥ 5”. We also removed lines 656-60 “An in-silico data search showed that nine of them, Rxrg, Fosl2, Pml, Runx2, Egr1, Ddit3 (Chop), Tfap2a, Tfap2b, and Fosl1 encode transcription factors/regulators with at least partial nuclear localization and known functions in NC and/or Schwann cells (during normal development or repair processes), suggesting that their deregulation could affect Adar1cKO Schwann-cell differentiation in a cell-autonomous manner.”
- Lines 659-660 now 671-72 are removed.
-We also removed sentence lines 707-08 “By contrast, no deficiency in peripheral nerve myelination has been noted in two independent Egr1 knockout lines, including the one we used in this study (38, 56).” Because of its redundancy with results section.
Comment 9. Paragraph from lines 622-639 is redundant, not necessary and too long. If you need to cite here your previous work, please do it in just 1-2 lines.
- We removed lines now 632-638 “RTqPCR and RNA-seq experiments suggested that activation of the Mda5/Mavs pathway and IFN and ISG activation could underlie the pigmentation and myelination defects of Adar1cKO animals, and partial rescue of myelination defects was, indeed, shown in cell culture models (27). Because each of the steps of Schwann-cell development is under the control of transcriptional regulators that form well-defined regulatory networks (30, 33, 35, 63, 64), we speculated that the deregulation of some transcriptional regulator could play a role in the genesis of the Adar1cKO mutant peripheral nervous system phenotype”. We also removed lines now 644-651 “Among these, the expression of Sox10, Egr2, Egr3, yy1, Sox2, Jun, Id2, Pax3, and Nfkb1 were not significantly altered in the Adar1cKO mutants relative to controls. Analysis of an initial RNA-seq 1 dataset generated from the sciatic nerves of Adar1cKO mutants and controls at P4 led to the publication of a first list of 346 transcriptional regulators deregulated more than two-fold in mutants relative to controls (27). Because most were upregulated in the mutants, we speculated that aberrantly continued expression of one or more factors that normally decrease may underlie the myelination defects of Adar1cKO animals (27) “.
- As proposed by the reviewer, only 7 lines are left to describe previous and novel results. You can now read “Recently, we highlighted a role for Adar1 in two NC derivatives: melanocytes and Schwann cells (27). Because the coordinated expression of several transcription factors is known to increase, in particular, between the pro-myelinating and myelinating stage (the stage affected in Adar1cKO mutants), whereas others are known to decrease strongly (30, 32, 33, 35, 63-66) we speculated that the deregulation of some transcriptional regulator could play a role in the genesis of the Adar1cKO mutant peripheral nervous system phenotype. We therefore re-evaluated our previously published RNA-seq dataset along with a new one and thus identified 52 transcriptional regulators that were up- or down-regulated in the Adar1cKO mutants relative to the controls with a FC ≥ 5. “
Comment 10. Conclusion: Please, correct the capital letters.
Agree. Capital letters are corrected (lines 764 and 765)
Comment 11. The conclusion should be shorter and re-focused to give a concise “take home“-message. Please, remove the cites and references of this section (!!). Lines 751-754 should be moved to the discussion section.
Thank you for pointing this out. The conclusion is modified accordingly.
- Lines 751-757 and associated references have been reintegrated to the discussion section (now lines 759-63).
- The conclusion section was modified. You can now read “Our study highlights the contribution of a small number of transcription factors to the defects observed upon Adar1 deletion in the NC, and possibly links A-to-I post-transcriptional and transcriptional regulation to the myelination process of the peripheral nervous system. In particular, our results argue that absence of Adar1 in NC derivatives results in the failure to shut-off early developmental regulators among which is EGR1, the continued presence of which in differentiating Schwann cells interferes with the maturation and myelination process.”

Reviewer 3 Report
Comments and Suggestions for Authors
1- Line 219, should be "were" not was.
2- This is an excellent manuscript, very detailed, addressing an important topic.
3- However, there is an important problem: Figure 6.
4- There is no "Figure 6"
5- Creates a big problem. How does this affect the other figures?
6- This must be corrected.
Author Response
Response to reviewer 3 comments
Thank you very much for taking the time to review this manuscript. Please find the detailed responses below and the corresponding revisions/corrections in track changes in the re-submitted files
Point-by point response to comments and suggestions
Comment 1- Line 219, should be "were" not was.
We agree with this comment. Page 5 line 221 you can now read “were”
Comment 2 This is an excellent manuscript, very detailed, addressing an important topic. However, there is an important problem: Figure 6.There is no "Figure 6" Creates a big problem. How does this affect the other figures? This must be corrected.
The Editorial Office layout missed Figure 6, which has been added. Other figures are not impacted.

Round 2
Reviewer 1 Report
Comments and Suggestions for Authors
The authors have addressed all the issues raised previously.
Reviewer 3 Report
Comments and Suggestions for Authors
A- The response of the authors to my previous comments were clearly addressed. This makes the manuscript a significantly improved version.
B- Some examples, Line s 184,185; Figure 2, changes on page 12, page 16, particularly page 18, page 21; and others.
C- These changes elevate this manuscript to higher scientific level.